# From discouragement to self-empowerment. Insights from an ethnolinguistic vitality survey among the Kashubs in Poland

**Justyna Olko**[1]*, **Karolina Hansen**[2], **Michał Wypych**[2], **Olga Kuzawińska**[2], **Macéj Bańdur**[3]

**1** Center for Research and Practice in Cultural Continuity, Faculty of "Artes Liberales", University of Warsaw, Warsaw, Poland, **2** Faculty of Psychology, University of Warsaw, Warsaw, Poland, **3** Kaszëbskô Jednota, Gdynia, Poland

* jolko@al.uw.edu.pl

**Data Availability Statement:** The data have been attached to the submission as supplementary material.

## Abstract

The paper relates the results of an ethnolinguistic vitality (ELV) survey among the Kashubs in Poland. The results reveal two interrelated layers of ELV: (1) an individual ELV reflected in language use and shaped by personal experience, emotions, and language proficiency; (2) a more collective ELV associated with the perception of the group's language strength, its status and utility. The most surprising predictor of linguistic praxis in our study, in addition to language skills, was the positive impact of experienced discouragement on language use. This remained significant when controlling for proficiency. We argue that the correlation between experiencing discouragement and increased language use is best explained by the self-empowerment of speakers who, earlier in their lives, met with negative attitudes toward their heritage language. Rather than succumbing to this discouragement and assimilating to the dominant language, their response was to develop an emotional link to Kashubian and increase their use of this minority language as a conscious act of self-determination and engagement.

## Introduction

After years of neglecting, downgrading, denying and even punishing speakers of minority languages, in 2005 the Polish parliament adopted the "Law on the national and ethnic minorities and on the regional language". It recognized thirteen national and ethnic minorities and their languages within the territory of Poland, as well as one 'regional language'–Kashubian. From that time on, Kashubian has enjoyed the official status of the only 'regional language', as opposed to a 'dialect' or a 'corrupt variant of Polish', which had been common terms used for Kashubian previously. Some prophesized that this new law would lead Kashubs to claim some form of regional autonomy, while others saw it as an important step toward better protection of ethnic minorities' rights in Poland. Now, almost fifteen years later, we find the Kashubian language taught in public and private schools in the region. We also see bilingual road signs, and, as the language enjoys a greater public presence, there also appears to be more pride in

**Funding:** The research is funded by the Foundation for Polish Science within the TEAM Program. The funders had no role in study design, data collection and analysis, decision to publish, or preparation of the manuscript.

Kashubian identity. However, it is unusual to hear the language being spoken in public domains outside of the classroom, especially among young speakers. Despite apparent challenges related to the future of their language, Kashubs appear to be a relatively strong and active ethnic minority, both in Poland and in the broader European landscape. In the summer of 2018 we decided to launch an internet survey in the region of Kashubia, in close collaboration with local activists and the Kashubian media in order to assess the strength of the group's ethnolinguistic vitality (ELV) as well as its language-related ideologies and attitudes.

In a manifesto known as the Strategy for the Protection and Development of the Kashubian Language and Culture (2006), the Kashubs declared that "the Kashubian language constitutes the foundation of the identity of the Kashubs" [1 p142]. According to the 2011 official census, 232,500 Polish citizens declared Kashubian identity, 16,400 of whom listed this as their only identity. Less than half of these respondents, namely 108,100 people, declared using the Kashubian language at home. The nature of the census question, which focused on active speakers, suggests that the figures above may in fact underestimate the total number of Kashubian speakers. These results also indicate that modern Kashubian identity is complex and is not based solely on language, which leads to the conclusion that the term 'community using the regional/Kashubian language', coined by Polish lawmakers to describe the Kashubs, does not fully correspond to reality. In the 2011 census, Kashubs were the second biggest group to declare an identity other than Polish, preceded only by the officially unrecognised Silesians with 846,700 declarations, followed in third-place by Germans with 147,800 declarations [2]. Thus, the imposed label of a 'group using the regional language' ignores the aspirations of at least part of the Kashubian community, who desire the full status of an ethnic minority, even if the current 'tacit minority status' is acceptable for the majority of Kashubs [3 p133, 4 p.706-729].

In general, the 2005 law has become the basis for public policy with regard to ethnic groups in Poland. Despite unquestionable improvements in minority language policy, it has also imposed rigid top-down classifications on to some of the minorities. Moreover, it effectively excluded other groups who were striving for official recognition, which is arbitrarily assessed and granted according to the fulfillment of statutory criteria for membership in each category [3 p133]. In other words, the provisions of the law "also set the limits of the minorities' empowerment, constituting sites from which they can speak" and imposed certain "identity formulas" on to specific groups [5 p47-50]. Moreover, Polish authorities have been repeatedly criticized for their lack of full compliance with the provisions of the European Charter for Regional and Minority Languages and for failing to address specific recommendations. Examples of this refer to the lack of education in which minority or regional languages should be used as the media of instruction, the absence or limited availability of textbooks in minority/regional languages, a lack of teacher training and a limited media presence [6, 7]. The last several years have also seen an increase of nationalistic ideology in state politics, unfavorable for ethnic minorities and their languages. For example, an amendment to the 2005 law was passed by the parliament at the end of September 2015, but the president Andrzej Duda vetoed it immediately after parliamentary elections the following month. However, despite this adverse political context, the last decade witnessed "mobilization, self-organization and institutionalization of ethnic life", with numerous new associations and projects aiming to preserve the cultural-linguistic resources of minority groups. Increasingly, such groups have used the Internet (and particularly social media) to achieve their goals, building intra-group relations and solidarity [5 p18-19].

As it became immediately apparent, our survey brought to the surface some of the enduring and contentious topics of the public debate regarding Kashubian, including whether the Kashubs are objects of ethnic discrimination, and whether there is a separate Kashubian

nationality. Some of the components of our questionnaire provoked heated debates on social media and in the local press, particularly amongst non-Kashub residents of the region, who saw it as an attempt to demonstrate the presence of ethnic discrimination which in their opinion does not exist.

Our goal was not only to assess the ELV of the Kashubs, but also to explore how its different dimensions, such as actual language practices, relate to other variables, for example, experiences of discrimination and discouragement towards language use. In particular, we wanted to see how past experiences of ethnic discrimination and assimilationist language policy have influenced present generations of Kashubian speakers, and whether the relatively recent recognition of this minority language has helped to challenge negative language ideologies. On a broader level, our aim was to see how this specific, dynamic case study might contribute to a more general theory of the ethnolinguistic vitality of ethnic minorities. To this end, we developed an extensive tool for assessing ELV, drawing on a number of previous approaches but also introducing new measures that combine perspectives from social psychology and sociolinguistics. As discussed in more detail below, previous measures of ELV, while having many merits, nonetheless suffer from problems from a statistical point of view and from their relatively narrow disciplinary focus. Our goal was not only to develop a current and multidimensional diagnosis of ELV, but also to better understand the key relationships and processes that determine ELV, including the actual language praxis of a minority group. To achieve this, we needed a tool that would be sufficiently statistically robust to allow for such analyses.

In the next section of this paper we present the historical, sociopolitical, sociolinguistic and cultural background of the Kashubs, providing a necessary context for our study. This is followed by an outline of the theoretical context, with a special focus on ethnolinguistic vitality theory and the approach taken in this study. We then provide an overview of the participants of our survey, its methods and the measures used, continuing with the presentation of its results including descriptive statistics and analyses focusing on the variables that predict the use of Kashubian and explain 'the layers' of ethnolinguistic vitality of this group. Subsequently, we test the effect of discouragement on the use of Kashubian as mediated by positive emotions experienced when speaking. We also examine an alternative model in which minority language use is a mediator of the effect of discouragement on positive emotions experienced while speaking. Finally, we discuss and interpret the results, drawing on the conceptual frameworks of empowerment and the Rejection-Identification model.

## Context of the study

**Kashubs** (*Kaszëbi*) are an ethnic group inhabiting the historical region of Pomerenia (Kashubia) in modern north-central Poland. According to the 2011 census, 233,000 inhabitants of the region declared their ethnic-national identity as Kashubian, while over 16,337 chose an exclusive Kashubian nationality. They are traditionally speakers of Kashubian (*kaszëbsczi jãzëk*), which is a West Slavic language influenced by Polish, Low German and High German [8]. Kashubian belongs to the same language family as Polish and for a long time was perceived as a dialect of the Polish language or as 'broken Polish', which contributed to its low prestige. There is considerable regional variation in Kashubian, including an entire continuum of forms between Kashubian and local Polish.

Both Kashubian identity and language use have been strongly influenced by the group's history. This includes a long period of Germanization between 1772 and 1918, when Kashubian territory was integrated into the Kingdom of Prussia and then remained under the rule of the German state, except for an eastern portion that was incorporated into the re-established Polish state (not including the area of the Free City of Gdańsk). During World War II the majority

of Kashubs faced compulsory enrollment in the Deutsche Volkliste and after 1945 were subject to the verification of nationality and subsequent rehabilitation by the Polish state, which was a complex process often accompanied by harsh persecutions [4 p653-655]. In the postwar period Poland was transformed from a multiethnic, multicultural and multilingual state into a linguistically-unified country that carried out an aggressive policy of assimilating ethnolinguistic minorities. As aptly summarized by Wicherkiewicz, "After the shift of political borders to the west, and a dramatic change in its ethnic composition, Poland was proclaimed a (quasi-) mono-ethnic (and as a result a monolingual) state" [1 p145]. For the Kashubs, this post-war policy in the period from 1945 to 1989, in which minority communities were reduced to folk-loric ethnic groups within the heritage of the Polish nation, resulted in the delegitimization of their language [3 p101-104, 9]. This ban extended to the use of the term 'language', which was replaced with 'Kashubianness' and 'Kashubian speech'. Children and youth who spoke the heritage language were mocked, punished and stigmatized as uneducated peasants. For these youth, the only way to get rid of this stigma was to merge with the upper class of speakers of the standard national language, who were typically perceived as 'better' [4 p667, 10 p61-65, 11, 12 p140-141, 13]. This repressive policy of stigmatisation provoked much resistance among the Kashubs and contributed to cementing their group identity, although this was not enough to stop the process of language loss from accelerating [14 p97].

While the situation of minority speakers in Poland improved after 1989, Kashubian was only officially recognized as a regional language of Poland in 2005. The process leading to the recognition of Kashubian as a regional language was supported by a number of activities in the field of language policy, including strengthening the presence of the language in public space, media and religious life. As discourse regarding Kashubian changed from framing it as a dialect of Polish to recognizing the linguistic distinctiveness of the Kashubs, the prestige of Kashubian started to rise [1]. While the 2002 census recorded over 52,000 persons who reported using Kashubian at home, as already mentioned, this number grew to over 108,100 in 2011, with approximately 3800 declaring it to be the only language used in this context. This may, however, reflect a growing willingness to admit a Kashubian identity and knowledge of the language, rather than an actual growth in the number of speakers of the language. A further important factor that needs to be considered is the distinct methodology of these two censuses; for example the 2002 questions clearly favored answers highlighting the use of Polish [15, 16 p242-243]. While there may indeed be as many as 100,000 active speakers nowadays, there are approximately several tens of thousands of people who know Kashubian but rarely speak it [10 p31].

There are many types of Kashubian speakers, who can be distinguished by the ways in which they use the language and according to their proficiency in it [cf. 17, 18]. There are those who use it only in specific domains and exclusively in an oral form, those who speak Kashubian-influenced Polish and view it as their mother tongue, persons who learned literary or standardized Kashubian, but rarely speak it and, finally, new speakers who have made Kashubian the main language of their daily life [10, 19]. The main demographic of Kashubian speakers comprise elderly and middle-aged native speakers or semi-speakers who have never had any Kashubian education and use the language in speech, but rarely or never read in it. There is also a vast group of middle-aged, adult, and adolescent semi-speakers and passive speakers, who have some linguistic competence due to their Kashubian-speaking environment (usually grandparents, parents, neighbors or friends), but do not speak it fluently or prefer to use Polish. Thirdly, there is a group of young adults, adolescents, and children who are semi-speakers or passive speakers, many of whom did not originally have any knowledge of Kashubian, but who received a Kashubian education at school and have at least a very basic competence in speaking, writing, and reading. Fourthly, there is a small yet very active group of

activists and language enthusiasts of various ages who have decided to actively use Kashubian, although their command of the language ranges from fluent to very limited. This group is the most diverse as it includes native speakers as well as those who are self-taught, with no previous knowledge of the language.

Intergenerational transmission has seriously weakened from the second half of the twentieth century, resulting in a gradual shift away from Kashubian as the language of daily communication to that of a marker of ethnic identity [20] This also means that the adults and youths who speak the language today have usually made a conscious decision to use Kashubian, even if in the majority of cases they were not socialized in it at home (see further discussion of the language transmission, learning and domains of use in the analytical part). Significantly, in the 2011 survey, 13,800 respondents replied that their native/mother tongue (the first language learned) was Kashubian—94,300 fewer than those who declared that it was the language spoken at home [3 p149].

Despite the fact that Kashubian was recently given the official status of a regional language, negative language ideologies that were pervasive in the second half of the twentieth century still persist, including the association of Kashubian with backwardness and peasantry [10 p15, 14 p110, 21, 22 p22-23]. However, the official recognition of Kashubian as the regional language, as well as a large number of new educational and cultural initiatives, have contributed to a slow change in perceptions towards the language, as well as the emergence of committed groups of Kashubian activists, particularly visible in social media. Official recognition made it possible to create the literary standard of Kashubian that is now subject to all aspects of language planning. It also assured its permanent presence in the media, public events and the linguistic landscape generally, exposing more and more people to a positive image of the language [1]. The 2005 law also facilitated a significant expansion of the presence of the language at schools. Although Kashubian had already begun to be taught in schools in 1992 on a limited scale, it is now much more present in public education (ca. 450 schools and 20,000 students), particularly in primary schools [3 p152]. However, it is being mainly taught as a second/foreign language, mainly to Polish-speaking monoglots, so it is debatable whether the current model of Kashubian education will be able to sustain language use in the years to come [see e.g. 12 p145-149]. Moreover, increasing literacy in Kashubian, achieved also through Kashubian education, has helped to pave the way for the language to enter social media. Although limited, the use of Kashubian on-line has been steady, especially on Facebook, where Kashubian or Kashubian-Polish fan pages have attracted up to ca. 25,000 followers. It can also be heard on public radio and television, as well as in the Catholic Church, to a lesser extent. Moreover, Kashubian has started to be used as an auxiliary language in local administrative offices in five districts [3 p155]. It is also widely used on road signs, signboards, monuments and, sometimes in advertising, although oftentimes these instances represent a somewhat narrow, symbolic or solely visual presence of the language. Whereas sociological research carried out in the 1990s revealed that individuals with higher levels of education who aspired for social advancement usually abandoned Kashubian [11], more recently Kashubian has been consciously adopted by educated and ambitious Kashubs [10 p37]. As we will argue below, the mechanism behind this process can be better understood through the results of our ELV survey.

## Theoretical framework

The present article builds on several theoretical approaches to studying the sociolinguistic situation of minority groups and the processes of language shift and loss. In addition, we attend to acculturation and interethnic relations, including attitudes of minority members towards the

dominant group and the national language. The most relevant theoretical development for our study is *ethnolinguistic vitality theory* [ELV; 23–25, 27]. This theory aims at understanding and predicting the possible outcomes of a group's vitality in the context of minority-majority relationships. It does so be examining the broader context of inter-cultural and inter-group relations, with special attention paid to the role of the heritage language as a component of identity. The original theory has been referenced and refined in hundreds of publications in social psychology, communication research, sociolinguistics and anthropology, heritage language and variationist studies, as well as in the field of language acquisition.

In the original definition provided by Giles, Bourhis & Taylor [23 p308], ethnolinguistic vitality was related to a group's ability to behave as "a distinctive and active collective entity in intergroup situations." It was assumed that the more vitality an ethnolinguistic group had, the more likely it would be able to survive and thrive as a collective entity in the intergroup context. Originally, three groups of factors indicative of a group's ethnolinguistic vitality were proposed [24, 28]: (1) Status: economic standing, political power, linguistic prestige; (2) Demographic: absolute population numbers, birth rate, geographic concentration; (3) Institutional support: recognition of the group/language in the media, education in the group's language, the legal status of the language according to the government. This became the proposed *objective* framework of ELV theory that was subsequently enriched by measures of *subjective* (or perceived) vitality, referring to "group member's subjective perceptions of their own ethnolinguistic vitality" that "may help account for group member's intergroup attitudes, skills and motivations" [29 p147] and "mediate their intergroup behaviors" [30 p256]. The subjective component can actually predict vitality more, given that individuals act upon their perceptions and beliefs and are driven by cognitive and motivational factors [26 p175-177, 31]. As argued by Ehala [32] a group's ability to act collectively may be independent of its objective vitality, for example, in the case of "national awakenings" that may well occur despite long-term decline in the objective vitality of their groups.

The subjective vitality questionnaire (SVQ) was originally developed in order to measure group members' assessments of in-group/out-group vitality on the basis of each of the dimensions included in the objective vitality framework: status, institutional support and demographic factors. However, subsequent research has shown that the three components of objective vitality are not reflected as distinct components in subjective perceptions of vitality; rather the SVQ measures a single factor that could be defined as *the perceived strength of the group* [33]. Recent studies have also pointed out serious difficulties associated with measuring institutional support and status factors objectively, as well as achieving meaningful results for comparative research [27].

Extensions and revisions of the ELV theory link the assessment of vitality to communicative behaviors, attitudes and relational strategies during intra- and inter-group encounters [26 p179-181]. Communicative behaviors and related attitudes are of particular importance for our own approach, which builds on the predictive power of 'egocentric' beliefs about ethnolinguistic vitality, but also measures language attitudes and language use [34–36]. Our approach has also been inspired by the proposed link between ELV and acculturation theory, refining the understanding of the former as the group's ability to maintain language practices and ethnic identity while resisting assimilation, or, alternatively, fostering assimilation in cases of low ELV [37]. This corresponds well to a recent definition of ELV proposed by Ehala [38] as "a group's ability to maintain and protect its existence in time as a collective entity". In our understanding, ELV refers to the ethnolinguistic conditions that influence group language choices and behaviors and is, in turn, influenced by these choices and behaviours. It also embraces the group's ability and opportunities to continue using its ancestral or heritage language in the future.

Our proposed approach to assessing ELV also attends to internalized components of language ideologies [cf. 39], self-reported language attitudes (e.g. language vs. dialect), language choices (to use a minority vs. majority language in a given context), domains of use and proficiency. Language attitudes here refer to both individual and collective beliefs, feelings, preferences and desires about a specific language [40–42] that "can have a profound effect on the socio-political position of a language or a variety of language in a society", and, consequently, its "maintenance and transmission" [43 p 153–154]. As part of our assessment of ELV, we also adapted and developed tools for the assessment of language transmission, domains of language use, forms of discrimination as well as emotional aspects of language use.

The new set of tools we developed and applied to the Kashubian context draw on the strengths of previous approaches to ELV, but also attempts to better integrate socio-psychological and sociolinguistic frameworks for assessing and diagnosing the processes and mechanisms of language maintenance, use and loss. Our proposed theoretical and methodological approach helps to explain the current socio-linguistic situation of Kashubian, including the language attitudes of speakers and the group's ability to maintain its distinctive identity, as rooted in their heritage language. Moreover, the approach enables us to present a refined multifactorial model of the ELV of speakers of endangered languages.

In our study we aimed to explore a range of potential variables that could predict the domains and frequency of minority language use and the perceived strength of a minority group and its language. One obvious factor related to using one's language is linguistic proficiency. In other words, people who speak Kashubian more fluently would be expected to use it more often. However, other factors can also influence language use as well as the perceived strength of the group and its language. We also hypothesized that personal experiences, such as emotions felt while speaking the heritage language, or experiences of mistreatment or violence, may also impact language use and the perceived strength of the group and its language. In addition, we assumed that factors relating to beliefs about a minority language, such as convictions about it status (whether it is a language or a dialect) and its perceived utility on the labor market, may also be important predictors of language use and perceived group strength. Additional variables included in our ELV model relate to other significant factors, such as experiences of discouragement with regard to using the minority language that may influence the language attitudes and practices of the group. Being discouraged from using one's language, for example, by teachers or parents, is a different phenomenon from experiencing violence or mistreatment and may therefore relate differently to linguistic practices. According to the Rejection-Identification model, perceiving discrimination can strengthen ingroup identification [44] and may lead to more positive emotions such as pride related to one's identity [45]. Hypothesizing that, in the case of linguistic minorities, this process may be reflected in language practices, our research questions address whether being discouraged from using the heritage language may have similar results and be associated with similar emotions.

## Method

### Participants

The research was approved by the Ethical Commissions of the Faculty of "Artes Liberales" and of the Faculty of Psychology of the University of Warsaw. The design of the survey was submitted for consultation with members of the Kashubian community and participation was voluntary. The participants of the study were recruited primarily from activist groups interested in the local Kashubian culture and their shared Kashubian identity. The invitation for the survey was disseminated through social media by Radio Kaszëbë, the Association of People of Kashubian Nationality *Kaszëbskô Jednota* and the journals *Skra* and *Pomerania*. The study link was

accessed by 332 people. As participants dropped out of the study at different points in time, the *N* varies for different measures between 237 and 175. We aimed at recruiting people who had at least partial skills in Kashubian. As the study was available online, some people ($n = 21$) who participated did not fulfill this linguistic criterion and were subsequently excluded from the analyses. The final sample for all measures consisted of 154 participants ($M_{age} = 35.05$, $SD = 13.72$, 43.5% women, 55.8% men, 0.6% other).

Regarding the place of residence, 48% of participants were living in villages, 25% in small towns, and 27% in larger cities. Participants' level of education was high: 53% had attended higher education (compared to the country average of 30% according to Eurostat). While 98% declared Polish citizenship, 94% declared Kashubian nationality. In the total group of respondents who completed the survey ($n = 154$), 51% declared both nationalities, while 7% declared an exclusively Polish nationality, and 43% declared an exclusively Kashubian nationality. The 43% of participants stating a sole Kashubian nationality seems very high considering the figures presented in the 2011 state survey, where only 7% of respondents from the Kashubian region declared an exclusive Kashubian identity. In contrast to the 2002 census, the census of 2011 allowed participants to select a dual Kashubian and Polish nationality, which is a common choice for the majority of Kashubs; despite identifying with their own ethnic group, they also recognize their Polish nationality and do not see this as conflicting with their Kashubian identification [15 p23-25, 16]. However, the last two decades have also seen a growing advocacy for an exclusive Kashubian nationality [15 p25]. The numbers stated in the official census and in our survey are of course not comparable, but the high percentage of participants who declared an exclusive Kashubian nationality suggests that our sample is generally representative of the most active and ethnically-engaged members of the Kashubian community, rather than being representative of the entire region. Notably, the Kashubian region as a whole faced strong assimilationist pressure during the interwar period and especially after World War II. It has also experienced a strong influx of people from neighboring areas and other parts of Poland. Therefore, we consider first, that our survey results are more representative of the most active and ethnically, as well as linguistically aware groups of Kashubs, and second, that they do not reflect the broader picture of the whole region.

## Procedure and measures

The study was conducted online in Polish and an anonymous link was distributed in cooperation with local activists. The participants were informed that the questionnaire would ask about various aspects of language use and perceptions of Kashubian as a minority language. After providing their informed consent and answering demographic questions, they responded to various scales related to linguistic vitality that we describe in detail below (given that the survey was part of a larger project with different lines of research, other scales were also included). There was no compensation for the study, but participants were thanked and provided with information about the project and its potential to improve the situation for speakers of Kashubian and other minority languages in Poland.

In our study we employed the following scales:

**1. Subjective ethnolinguistic vitality:** Here we combined two separate scales: perceived group strength and perceive language strength. Regarding **group strength,** Bourhis, Giles & Rosenthal [29] presented the first attempt to measure subjective vitality in what they referred to as the Subjective Vitality Questionnaire (SVQ). Out of their original 22 questions we selected 7 items relating to status (economic standing, political power, recognition, respect, the perceived strength of their position now and within 20 years); these correspond to questions 6, 8, 13, 18, 19, 20 and 21 in the original SVQ ($\alpha = .77$). We selected several items that, in our

opinion, related most to the group's perceived strength, status and power in relationship to the dominant group. The complete original scale, when used previously in larger surveys, did not reveal the original three-component structure of the objective vitality questionnaire, comprised of status, demography, and institutional support factors [33]. As for the perceived **language strength,** we applied a separate scale based on three questions about perceived language status, the degree of recognition the language now receives and what participants believed the situation would be like in 20 years time (3 items, α = .76). This draws on the original items of SVQ [29] referring to language prestige and presence in different spheres of public life and education. However, we rephrased the items to provide a meaningful component corresponding to the abbreviated scale of group strength used in the survey. During analysis, we combined both scales into one measure reflecting subjective ethnolinguistic vitality (SBJ ELV, 9 questions, α = .80)

**2. Language transmission:** We developed this question based on sociolinguistic data regarding the most common situations of transmission for minority endangered languages, that is, learning not only from parents, but also grandparents and other relatives or outside the home. We also took into account the current sociolinguistic situation of the Kashubian language. Participants answered a multiple-choice question concerning the acquisition/learning of Kashubian. The question was phrased as follows: "Where have you learnt Kashubian? (You can select more than one option)".

**3. Language use**: One approach to assessing ethnolinguistic vitality is based on directly measuring language use. Our scale, similar to those used in previous studies, aimed to reflect a broad range of domains of language use, taking into account use of both the minority and dominant language in family circles, immediate social networks, with friends, in schools, institutions, services, public events, and on social media (12 items, α = .93). Previous scales of this kind include those by Allard & Landry [36], Ehala & Zabrodskaja [46] and the EuLaViBar Project [47]. In comparison with previous tools, we addressed frequent patterns of interrupted intergenerational language transmission that often skip the parents' generation, but are based rather on the oldest family members as well as other relatives who transmit the language to the youngest generations. Accounting for the modern Kashubian context, we also included places of worship, medical services, and social media; in recent years these have become important spaces for the use of minority languages worldwide.

**4. Discouragement of language use:** With this scale we measured participants' experiences of being discouraged from using the language by people in positions of control (e.g., parents, teachers, institutions), both during childhood and in the present day. This scale was based on our experience of conducting sociolinguistic research on minority languages, consultations with minority speakers, as well as other previous studies on the mechanisms of language loss and language attitudes [e.g. 48, 49]. It was also inspired by two items from the EuLaViBar Project: Q22 and Q24 [47]. In the final analysis we decided to exclude the question about discouragement from parents because of a poor item-total correlation (alpha increased from .65 to .72 after its removal). This decision reflects the small probability of this kind of situation being significant compared to other channels of discouragement. Moreover, we considered that participants might be unwilling to reveal discriminatory attitudes of their own parents.

**5. Experienced language discrimination:** This 3-item scale reflects situations of language discrimination in everyday life. For exploratory reasons, we applied all items for the past and present; however, our main focus in this paper was past discrimination (α = .80).

**6. Language use and emotions**: We developed this scale to measure the positive and negative emotions experienced by users of heritage/minority languages (e.g. shame, pride, guilt, joy, nostalgia). This component builds on earlier research pointing to the importance of the emotional dimension of ELV and indicating that emotional attachment to group identity

positively correlates to the level of ethnocentrism and, moreover, mobilizes members for collective action [46, 50, 51 p304-308]. In earlier research emotional attachment was measured through a self-reported scale regarding participant's tendency to value cultural heritage and tradition and was included in the six-factor Web Model of intergroup settings. However, we decided to take a different approach, measuring the frequency of both negative (4 items, $\alpha$ = .65) and positive (6 items, $\alpha$ = .87) emotions directly associated with the use of the minority language.

**7. Language ideologies concerning the status of Kashubian:** We asked two questions to measure language ideologies (e.g. the status of Kashubian as a language or a dialect, 2 items, $\alpha$ = .56, $r$ = .39).

**8. Economic utility**: A scale was employed to measure beliefs about the utility of the minority language in the labour market. This was adapted from the EuLaViBar project Q52 [47] to reflect the perceived economic value of speaking the minority language (4 items, $\alpha$ = .87).

**9. Self-evaluation of language proficiency**: We applied three questions to measure self-assessed proficiency in both Kashubian and Polish, based on the simple model developed by the EuLaViBar Project (Q28 and Q30). We added another simple 2-question component assessing the perceived difficulty of language use, as previously developed by one of the authors during sociolinguistic fieldwork with Nahua/Indigenous Mexican immigrants in the US.

## Results

Presented below (Figs 1–4 and Table 1) are descriptive statistics of the most important scales related to language acquisition, language use in different domains, language-related ideologies and discrimination.

The data show that the most frequently reported source for learning the language was the family: participants learned Kashubian mainly from their parents (34% from the mother, 36% from the father), and from grandparents as well as other members of the family (48%). Friends were the third most frequent source of language learning (30%). Official channels of language learning, such as school, university and language courses, account for 13%, 2% and 6% of language knowledge respectively. Other frequently cited modes of language acquisition, in response to our open-ended question, included radio, books, the Internet, and self-education. It is important to remember that participants could choose more than one answer, and the vast majority reported several sources of language acquisition.

The self-reported levels of proficiency in Polish and Kashubian varied across the sample. On a 1–5 scale, the mean value of proficiency in Polish was 4.71 (SD = 0.42), while for Kashubian it was 3.12 (SD = 1.14). In the self-assessment of language proficiency, the majority of

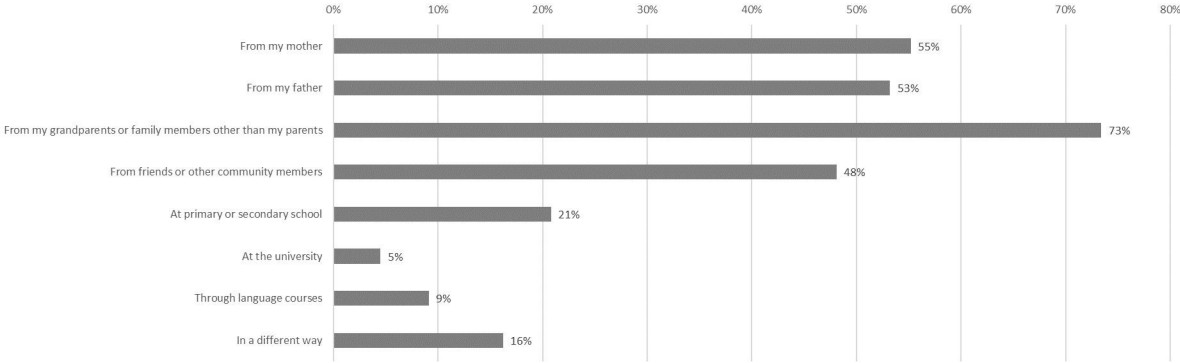

**Fig 1. Channels of language acquisition (in percent).**

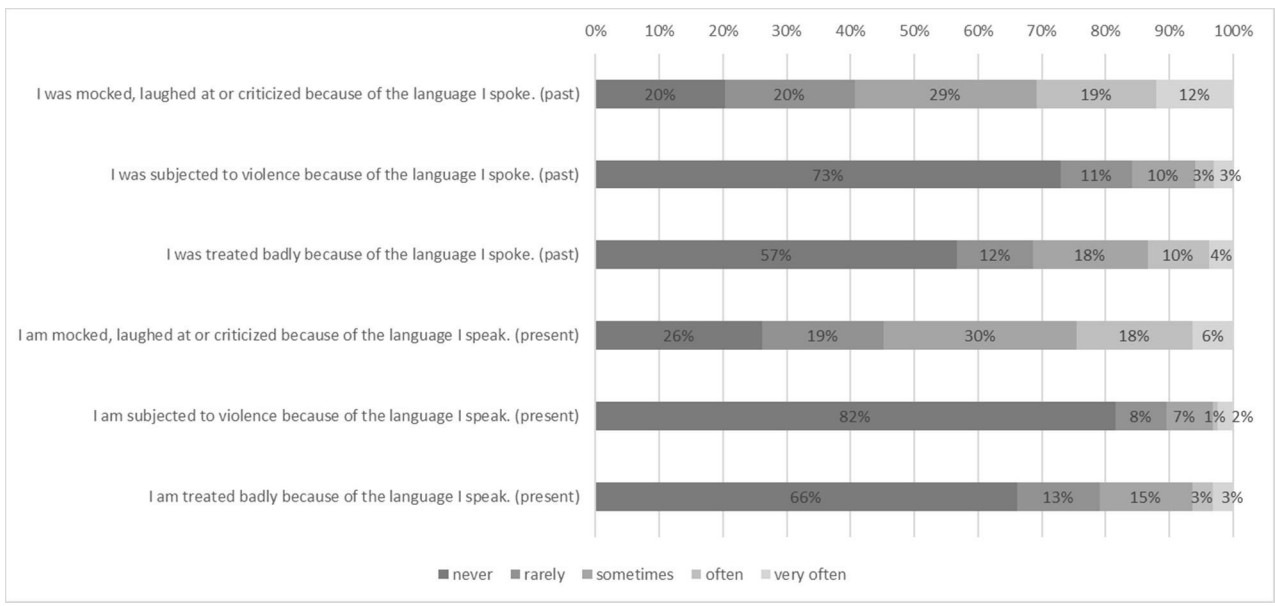

**Fig 2. Experience of discrimination and language avoidance in the past (in percent).**

participants (53%) declared that they could speak Kashubian well or very well. 21% indicated that they could speak the language "neither a little nor well". 18.2% stated that they knew Kashubian a little, whereas 7.1% indicated that they knew very little. With regard to the perceived difficulty of speaking Kashubian, similar results were obtained: 46% of respondents indicated that speaking Kashubian was easy or very easy, 27% stated that it was neither easy nor difficult and 27% declared that speaking the language was difficult or very difficult. However, almost half of the participants (over 48%) declared that they could write Kashubian only a little or very little. Over 30% of respondents declared that they could write well or very well in Kashubian. These results point to a significant difference between oral and written competence in Kashubian among respondents; they also show a broad range of language proficiencies, revealing several different kinds of speakers, from relatively fluent ones to possible semi- and passive speakers.

With regard to language use in the analyzed sample, 30% of participants reported communicating with their father in Kashubian, 47% in Polish and 23% in both languages. 26% of

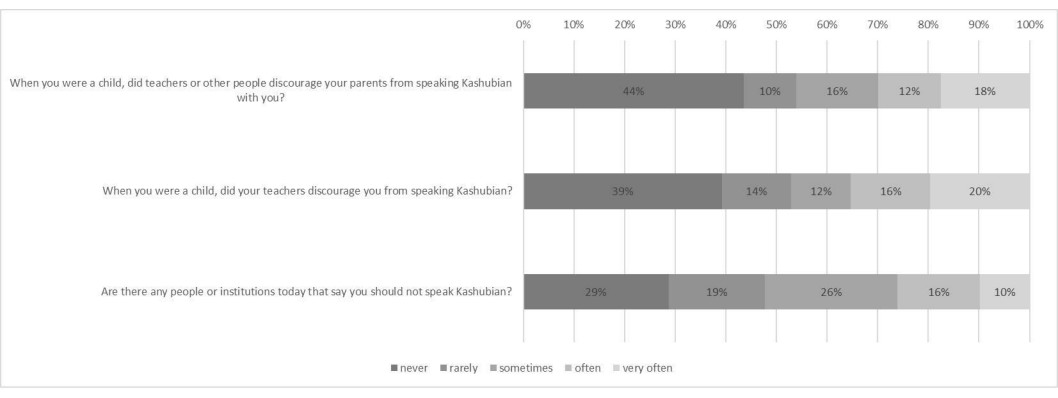

**Fig 3. Discouragement from speaking Kashubian (in percent).**

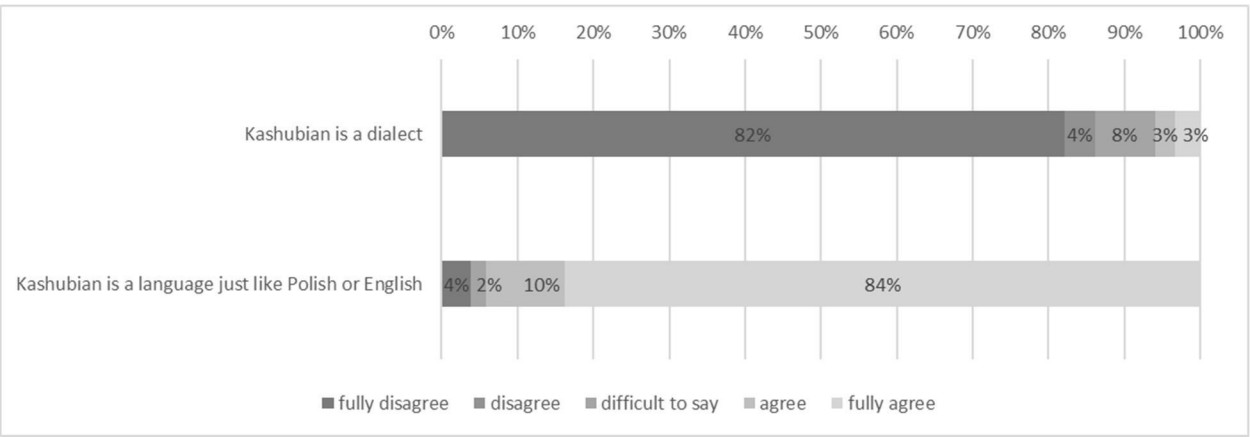

**Fig 4. Elements of language ideologies reported with regard to Kashubian (in percent).**

participants reported communicating with their mother in Kashubian, 49% in Polish and 25% in both languages. Regarding the use of both languages in everyday life, the most Polish-dominated linguistic domains are local authority contexts and health services: 93% of participants reported speaking Polish with authorities and 95% in health services. The domestic domain, on the other hand, is the one where more respondents spoke Kashubian. A total of 44% of respondents spoke more Kashubian than Polish with their grandparents (28% with parents). At the same time, only 13% spoke more Kashubian than Polish with their own children, which testifies to a serious gap in inter-generational transmission of the language. While cultural events and social media are platforms which might be expected to facilitate the use of Kashubian, this turned out to be true only to a certain extent: 12% and 9% of participants admitted speaking more Kashubian than Polish in these respective domains. The church remains almost exclusively a Polish-speaking space: a total of 4% reported speaking more Kashubian than Polish in church, while 9% reported speaking Polish and Kashubian to the same extent. It is important to add, however, that few masses are celebrated in Kashubian, so the language appears to be used to a limited extent in communication between speakers attending church.

When it comes to language attitudes, the most commonly reported behaviors were avoiding speaking Kashubian in the presence of Poles at work, at parties and in public spaces such as on public transport (40% and 31% of respondents respectively admitted to doing this "often" or "very often"). Less frequently reported were experiences of discrimination, such as violence and mistreatment related to the use of Kashubian: 7%-14% reported having experienced this regularly. Mockery, on the other hand, was experienced often by 29% of participants.

**Table 1. Domains of language use (in percent).** The following categories: 1 (only Kashubian), 2 (mainly Kashubian) and 3 (more Kashubian than Polish) were collapsed into Kashubian, whereas 5 (more Polish than Kashubian), 6 (mainly Polish) and 7 (only Polish) into Polish.

| Language type | | Subdomains of language use | | | | | | | | | | |
|---|---|---|---|---|---|---|---|---|---|---|---|---|
| | Parents | Grand parents | Children | Friends | Neighbors | At school or at work | In shops and services | With local authorities or in offices | At the doctor's | At cultural or community events | At church/ places of worship | On the Internet or in social media |
| Polish | 57 | 42 | 66 | 68 | 73 | 80 | 89 | 93 | 95 | 76 | 88 | 76 |
| Equal Polish & Kashubian | 15 | 14 | 22 | 21 | 14 | 12 | 7 | 3 | 3 | 12 | 9 | 15 |
| Kashubian | 28 | 44 | 13 | 11 | 13 | 8 | 4 | 3 | 1 | 12 | 4 | 9 |

Regarding discouragement from using the language, 57 to 61% of participants admitted that at some point in their lives individuals had discouraged them or their parents from speaking Kashubian; for 31 to 36% this had happened often or very often. When asked about the present situation, 70% were convinced that some institutions still advocated against speaking Kashubian.

Finally, when asked about ideologies related to language, 80% of respondents declared that Kashubian is a language, not a dialect.

## Correlations between scales

We conducted a correlational analysis in order to explore the relationships between proposed components of ethnolinguistic vitality, including language use and other variables.

The results (Table 2) show that subjective ethnolinguistic vitality (i.e. measured as perceived group and language strength) is related to the use of the minority language in different domains and situations. People who experienced more discouragement (or, more precisely, those who perceived and declared having experienced more discouragement) declared higher minority language proficiency and more frequent use of the minority language across different domains. Furthermore, these respondents also experienced more positive emotions when speaking Kashubian. The pattern described here is distinct from when participants reported their past experiences of ethnic discrimination: people who experienced more discrimination in the past perceived lower levels of vitality in relation to both their group and language.

Moreover, these participants declared both positive and negative emotions when speaking Kashubian. However, the experience of discrimination is not correlated in any significant way with use of the minority language. Experiences of discrimination and discouragement are of course correlated [.59***]: people who were discouraged most from speaking the language, also reported more experiences of discrimination. Looking at positive emotions, in addition to their surprising relationship with discouragement and discrimination, they were also related to the use of Kashubian in different domains and to a higher proficiency in the language. Negative emotions were associated with experiences of discrimination and were also correlated with lesser usage of the minority language across different domains. In turn, perceiving Kashubian as a language rather than a dialect was related to a higher perceived subjective ethnolinguistic vitality. This correlation is further supported by qualitative data on Kashubian activists,

**Table 2. Correlations among aspects of ethnolinguistic vitality and other variables.**

|  | Language use | Lang. profic. | Discouragement | Past experienced discrim. | Positive emotions | Negative emotions | Lang. status | Econ. utility |
|---|---|---|---|---|---|---|---|---|
| Subj. vitality | .31** | .06 | -.15 | -.24** | .09 | -.10 | .27** | .15 |
| Language use |  | .60** | .27** | .07 | .33** | -.27** | -.01 | .22** |
| Lang. proficiency |  |  | .21** | .11 | .20* | -.07 | -.09 | .35*** |
| Discouragement |  |  |  | .58** | .25** | .04 | -.08 | .02 |
| Past experienced discrimination |  |  |  |  | .22* | .17* | .01 | -.11 |
| Positive emotions |  |  |  |  |  | -.13 | -.15 | .09 |
| Negative emotions |  |  |  |  |  |  | -.03 | -.05 |
| Lang. status |  |  |  |  |  |  |  | -.18* |

* $p < .05$

** $p < .01$

*** $p < .001$.

who noticed a positive change in language attitudes and interest in speaking Kashubian after its recognition as a regional language [10 p83-84]. Surprisingly, however, respondents who regarded their language as being useful in a work setting and perceived it as having a higher economic value, tended to consider it a dialect rather than a language. Nonetheless, they also shared a higher subjective ethnolinguistic vitality and self-assessed proficiency, and they spoke Kashubian more often in different contexts.

## Layers of ethnolinguistic vitality

Our subsequent analysis embraced two different, though interrelated, measurements of ELV. On the one hand, we verified which of the variables would specifically predict the use of the minority language. On the other hand, we looked at a more general subjective assessment of ethnolinguistic vitality (SBJ ELV), as expressed through perceptions of group and language strength see [29]. Although we had certain expectations about which factors would likely influence different measures of vitality, in order to remain open to other relationships, we tested a broader set of factors in the analyses. In the two regression analyses, we included the self-assessment of minority language proficiency, negative and positive emotions while speaking, experiences of being discouraged from using Kashubian, perceived past language discrimination, language status, and perceived economic utility of Kashubian.

The results (Table 3, left) revealed that more frequent and extensive use of Kashubian was predicted to the largest extent by self-assessed proficiency in this language and a lack of negative emotions when speaking. Other significant predictors were positive emotions while speaking and experiences of being discouraged from using the language. Perceived past discrimination, language status and the perceived economic utility of Kashubian did not play a significant role in predicting its usage (taking at the same time all other factors into account).

The results for subjective ethnolinguistic vitality as a dependent variable (language and group strength; Table 3, right) showed that it was not predicted by minority language proficiency, negative or positive emotions, nor experiences of being discouraged from using the language. It was, however, predicted by perceived language status (perceiving Kashubian as a language rather than a dialect), the perceived economic value of Kashubian, and by experiences of discrimination. Although subjective ethnolinguistic vitality and economic utility did not appear to be significantly related when examining correlations (Table 2), when controlling for other variables (such as language proficiency) they were.

It seems, then, that two interrelated layers of ethnolinguistic vitality can be distinguished. The praxis-related ethnolinguistic vitality, as reflected in daily language use, depends on

**Table 3. Predictors of minority language use and subjective ethnolinguistic vitality.**

| Variable | B | SE (B) | β | B | SE (B) | β |
|---|---|---|---|---|---|---|
| | | Language use | | | Subjective ethnolinguistic vitality | |
| Language proficiency | .44 | .07 | .49*** | -.02 | .05 | -.04 |
| Negative emotions | -.43 | .14 | -.21** | .05 | .10 | .05 |
| Positive emotions | .15 | .08 | .13* | .09 | .06 | .12 |
| Discouragement | .12 | .07 | .14* | -.01 | .05 | -.03 |
| Past experienced discrimination | -.01 | .09 | -.03 | -.13 | .06 | -.23* |
| Language status | -.02 | .10 | -.01 | .27 | .07 | .31*** |
| Economic utility | .21 | .13 | .13 | .26 | .09 | .26* |

*$p < .05$.

***$p < .001$.

individual factors such as language proficiency, emotions and personal experiences of discouragement. The subjective ethnolinguistic vitality, reflected in the perceived group and language strength, appears to be related to more abstract and general perceptions of language status and utility, but is also influenced by experiences of discrimination. Thus, although discouragement and experiences of discrimination are correlated [.59***] they appear to act differently on the distinct facets of ethnolinguistic vitality.

## Discouragement and language use

As we can see in the regression, experiences of discouragement regarding the use of Kashubian had an unanticipated effect on participants' language use: the more they were (or perceived to have been) discouraged from using it, the more they speak the language. Further, as we can see in the correlations, discouragement was also related to positive emotions (but not to negative ones). In order to better understand the impact of discouragement on language use, we tested a model via a mediation analysis with bootstrapping (using Process macro for SPSS). We checked whether the effect of discouragement on minority language use was mediated by positive emotions when speaking the language. Given that language proficiency plays a crucial role in language praxis (as discussed above), and as we wanted to understand the possible effects beyond the impact of proficiency, we included it as a covariate in all the analyses.

The results show that the more participants had been discouraged from using the language, the more positive emotions they felt while speaking Kashubian. Furthermore, the more positive emotions they experienced, the more they used the language (Fig 5). Of course, language proficiency influenced language use and it also influenced the experience of positive emotions while speaking. While the mediation was not full, the indirect effect was significant (confidence intervals = CIs do not include zero). Thus, the relationship between higher levels of discouragement and higher levels of language use was partly due to the positive emotions that participants experienced when speaking the language, even though they had previously been discouraged from using it. This effect is observed even when controlling for language proficiency (i.e. making it statistically similar for all the participants). Thus, even though our

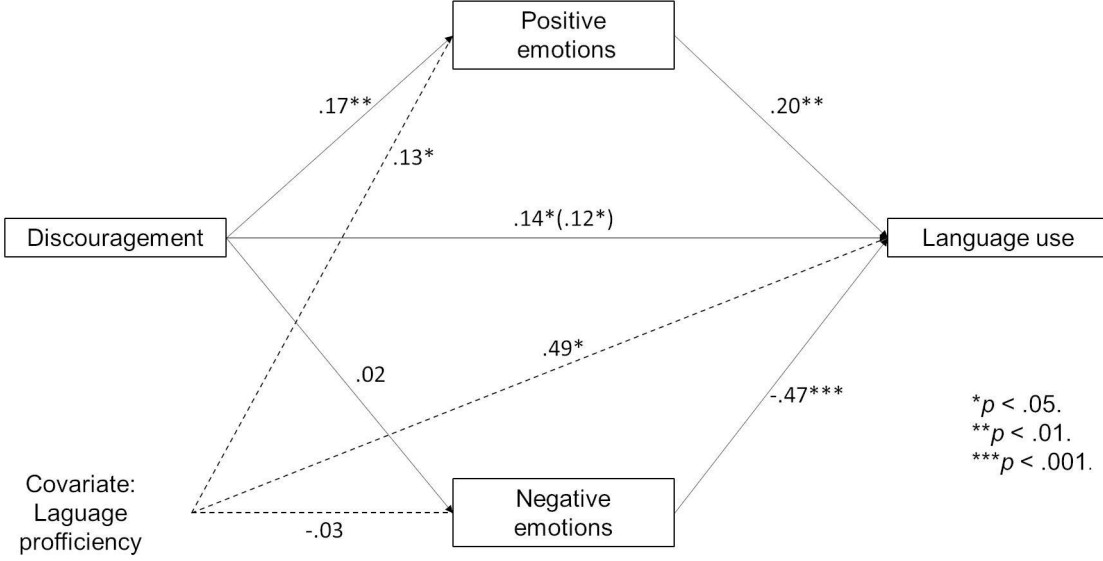

**Fig 5. Indirect effect of discouragement on the minority language use through positive emotions (while controlling for language proficiency).** *p < .05. ** p < .01. ***p < .001.

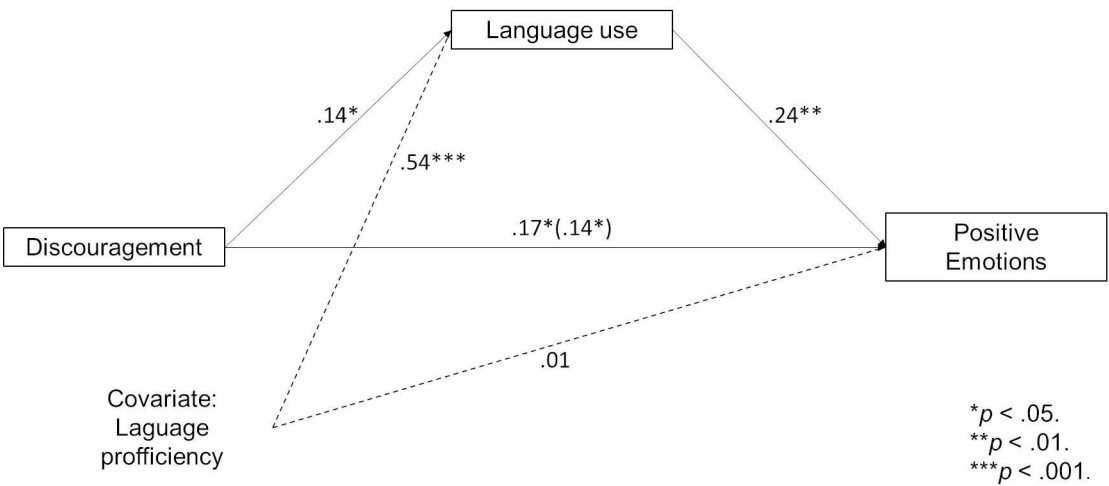

**Fig 6. Alternative-sequence model of discouragement influencing language use and emotions.**

participants differed in declared levels of the minority language competence, the experience of discouragement had similar effects for all of them. The relationships and indirect effects presented in this article are stronger when language proficiency is not taken into account; however, we prefer to present well-controlled and more cautious versions of the models. Notably, discouragement did not influence current negative emotions and we did not find an indirect effect as a result of negative emotions.

Theoretically, the processes described above could also be reversed. To explore this further, we tested an alternative model with language use as a mediator of the effect of discouragement on positive emotions experienced while speaking. We kept the self-assessed language proficiency as a covariate in the model. For discouragement, the results were similarly strong as in the earlier model testing the previously described order. They showed that the more participants were discouraged, the more they now use the minority language and the more positive emotions they feel while speaking (Fig 6). In sum, it is possible that experiences of discouragement initially influence the use of the Kashubian language and subsequently influence emotions linked to speaking it.

## Discussion and conclusions

In original approaches to ethnolinguistic vitality (ELV) its subjective dimension was linked to exocentric beliefs [29]. However, the extended model of subjective vitality not only included exocentric beliefs concerning the perception of vitality at present and in future, but also egocentric beliefs assessing participants' sense of belonging to the group and associated personal goals [34, 36]. The extended subjective vitality model was further elaborated by Harwood, Giles and Bourhis [26] through the inclusion of a vitality assessment aimed at addressing a broad range of communicative behaviours. However, the actual language practices of speakers were often absent from, or insufficiently explored, in these previous approaches to quantitatively measuring the ethnolinguistic vitality of a given group. Exceptions include questionnaires and approaches developed by Landry and Bourhis [52] and by the EuLaViBar Project [47]; the latter was based on a sociolinguistic approach and did not employ the tools developed in the field of social psychology within the ELV theoretical framework.

Our approach combined sociolinguistic and psychological perspectives, emphasizing the communicative level, that is, language praxis. The results of the survey carried out with the

Kashubian minority reveal two interrelated layers of ELV. First, an individual ELV reflected in language use and shaped by personal experiences, emotions, and language proficiency. Second, a more collective ELV associated with the perception of the group's strength, as well as the strength, status and utility of the language. The two dimensions are, of course, closely linked and should be seen as a complex, dynamic and interrelated process: the strength of the group and their language, as well as the perceived economic value of the minority language, correlate with its use across different domains.

However, the most surprising predictor of linguistic praxis in our study, in addition to language skills, was the impact of experiences of discouragement towards language use. This remained significant when controlling for proficiency. In a cross-sectional study such as this one we are not able to determine the direction of influence of variables with any certainty. Thus, even though it seems unlikely, we cannot rule out the possibility that people who are more competent in the minority language, use it more often and thus perceived more instances of discouragement than those who use it less. This scenario, however, seems less probable when we take into account the role of emotions in ELV. According to the Rejection-Identification model, perceived discrimination strengthens ingroup identification [44, 53]. As for the directionality of this relationship, longitudinal research has shown that perceptions of discrimination have the causal effect of increasing group identification among persons exposed to discrimination [54 p656]. High levels of ethnic identity in terms of a sense of belonging and commitment to one' s ethnic group seem to play a protective role in self-perceived worth as a person and also contributed to higher perceived academic performance [55]. Of particular relevance to our study is longitudinal research relating to Latino college students in the US, which has revealed that those who perceived more ethnic discrimination also identified more with their ethnic group. In turn, those with higher ethnic identification exhibited greater well-being and activism [56].

In our case, rather than experiences of ethnic discrimination per se, a similar effect is seen with regard to experiences of discouragement and external pressure to reject the heritage language. Rather than contributing to its abandonment, discouragement was associated with a greater use of Kashubian and positive emotions toward speaking it. Language practices in the case of minority ethnic groups are, of course, a strong indicator of personal and group identity, but the impact of discouragement is best seen in the frequency of language use within a range of domains. Thus, discouragement appears to be a key component of the ethnolinguistic vitality of a linguistic minority. The study therefore reveals a particular pattern amongst the Kashubian respondents who participated in the research; experiences of discouragement did not reduce language use, but in fact positively influenced it. In terms of sociolinguistic praxis, minority members often become committed activists and speakers of an endangered language when they decide to reject and resist imposed majority ideologies. As discussed in more detail below, this explanation is additionally supported by the specific profile of our survey participants, who were recruited as active members of the Kashubian minority and thus more likely represent a more engaged and active group than the broader community. Of course, in addition to experiences of discouragement, other factors may play an important role in predicting language use, for example: the professional profile of survey participants linked to Kashubian identity and heritage, membership in an activist group, or re-evaluation of the heritage language as a result of its formal legal recognition as a regional language in 2005. However, as we showed earlier, factors such as the status of the language do not appear to have a direct effect on language use (when we control for language proficiency).

We believe, therefore, that the results described in this study can be best explained by the conceptual framework of empowerment [57, 58], understood as achieving higher degrees of autonomy, self-control and self-determination in claiming one's rights as well as recognizing

and using available resources. It can refer to both individuals and groups who "become able to take control of their circumstances and achieve their own goals, thereby being able to work towards helping themselves and others to maximise the quality of their lives" [59 p8] and who focus "on ways to develop feelings of personal power and self-efficacy" [60 p816]. Previous studies on empowerment of ethnic minorities have mainly focused on educational contexts [e.g. 61–64]—including both the policies and practices of mainstream institutions and indigenous-run programs of language reclamation—as well as, to a lesser degree, in the area of law-making and language policy [e.g. 65]. Discussing the situation of Spanish speakers in the Southwestern United States, Martínez-Brawley & Zorita note that "the opportunity to assert one's language is extremely empowering" and it can be achieved without external validation of one's native culture and language [66 p90]. This perspective is reflected in the successful initiatives of a number of indigenous groups engaged in the process of reclaiming and revitalizing their heritage languages. For example, discussing specific programs and initiatives led by indigenous groups in the US, McCarty & Nicholas describe the processes of self-empowerment of Hopi educators who became "caretakers of the language" and creators of a school language policy based on educational sovereignty.

The key here is to understand the concept of empowerment not as "external intervention" or "bestowal of something by the strong" [67 p20], but as self-empowerment that is rooted in an active stance and explicit will of "traditionally disadvantaged minority groups" who can assume power, not just receive it [67 p24]. In his study of Native American experiences in the US educational system [characterized as "draining-out of hope";62 p. 139], William Tierney describes "rituals of empowerment" in teaching strategies through which minority students can gain a measure of control over their lives. Drawing on specific examples, he emphasizes that the key to such a process is to "create conditions where individuals may empower themselves" because "empowerment is not given; it is taken" [62 p149]. This perspective, emphasizing the agency of empowered individuals, or the process of self-empowerment—even when encouraged or facilitated by collective processes and other stimuli—is particularly suitable for the context of our study. Moreover, such processes are attested by group members who, as a result of different forms of social discrimination, have been excluded from decision-making processes, self-determination or other forms of control over their lives. Therefore, self-empowerment is anti-oppressive and closely linked to powerful forms of self or group advocacy amongst individuals or communities who have been marginalized or discriminated against [59 p12, 23].

As documented in earlier research and further confirmed in our survey, several generations of Kashubs have been subject to many forms of ethnic discrimination. Their situation was aptly described by Günter Grass in his famous novel "The Tin Drum": ". . .because we're not real Poles and we're not real Germans, and if you're a Kashube, you're not good enough for the Germans or the Polacks. They want everything full measure" [68 p416]. Our survey covered relatively recent experiences of discrimination, embracing to some extent the late communist period, but mostly focused on the post-communist era. It appeared that Kashub people's reactions to experiences of oppression differed over time, and included: acceptance of victimization, forced assimilation, adopting a conscious strategy to disassociate with the community and become a member of the dominant group, or resistance and resilience. In particular, it has been observed that more recently, Kashub attitudes towards their language have been shifting from a "resistance identity", formed during times of repression in the communist period, to a "project identity", focused on positive attitudes with regard to their ethnic belonging and opportunities for the group in the future [14 p116]. In other words, "Kashubs are now being empowered to choose how they self-identify, rather than having this identification imposed on them" [14 p111].

As already pointed out, the participants of our research were recruited through Kashubian media, including social media channels for individuals interested in Kashubian identity and language. Many of them were likely language activists or at least persons who were supportive of the language. As already mentioned, during our research, the Association of People of Kashubian Nationality *Kaszëbskô Jednota*, which connects Kashubs interested in their heritage and promotes Kashubian ethnic identity, agreed to share our survey via their social media accounts. Therefore, it is plausible that a large percentage of respondents could have sympathized with this particular association, especially given the very high percentage of people who declared an exclusively Kashubian nationality. The *Kaszëbskô Jednota* promotes the view that Kashubs form a nation on the basis of the self-identification of members of the community. This stands in contrast to the more dominant vision of Kashubian identity, which is maintained by the Kashubian-Pomeranian Association, whereby Kashubs are seen to constitute an ethnic or ethnic-regional group with ethno-regional, dual Polish-Kashubian identity [4 p681-715].

We argue, therefore, that our research reflects the experiences of active and engaged members of the Kashubian community, and that the association between experiencing discouragement and increased language use, as demonstrated in our survey, is best explained by the self-empowerment of speakers who had encountered negative attitudes toward their heritage language earlier in their lives. Rather than succumbing to assimilation, their response was to develop an emotional link to Kashubian and speak it more often. In this way, speaking the minority language comprises a conscious act of self-determination and presents an opportunity to reassess of the emotional and economic value of their language. Moreover, this process is often accompanied by emerging communities of practice, which become spaces for individuals to further develop and assert their ethnic identity. Thus, speaking Kashubian can be seen as a manifestation of a specific response towards the dominant culture and the pressure for linguistic unification. It is an expression of an "identity of resistance" amongst those who challenge their marginalized and stigmatized position in society by constructing a minority identity that stands in opposition to the dominant one [10 p206-210]. Furthermore, this identity reflects a long-term, positive and self-empowering personal and collective engagement. Through this engagement people counteract culture and/or language loss and make claims for both individual and group rights. Self-empowerment usually develops from a state of passivity into one of long-term commitment [60 p817], as is the case for many Kashubian groups who actively promote and advocate for the Kashubian language.

As our study revealed, the common mechanism in this process could have been a self-empowering reaction to experienced discouragement related to the use of the heritage language, and the discriminatory attitudes that speakers frequently encountered in schools and other public spaces. We propose that these experiences are in fact an important factor affecting the ethnolinguistic vitality of a minority group, directly affecting language praxis, which, in turn, is a key predictor of ethnolinguistic survival and continuity. Our study also has broader implications for language revitalization strategies, emphasizing the key role that both individual and group self-empowerment play in the process of language emancipation and in increasing the agency of minority speakers. It is self-empowerment that leads to achieving the most significant goal of language revitalization: speaking the endangered language.

## Supporting information

**S1 File.**
(SAV)

## Acknowledgments

We express our gratitude to the members of the Kashubian community who decided to take part in the survey and to the Kashubian media for sharing the link to the questionnaire. We also thank Bartłomiej Chromik, Nicole Dołowy-Rybińska and Tomasz Wicherkiewicz for their insightful comments on earlier drafts of this paper. This work has been developed within the project "Language as a cure: linguistic vitality as a tool for psychological well-being, health and economic sustainability" that is carried out within the Team programme of the Foundation for Polish Science co-financed by the European Union under the European Regional Development Fund.

## Author Contributions

**Conceptualization:** Justyna Olko, Karolina Hansen.

**Data curation:** Michał Wypych.

**Formal analysis:** Justyna Olko, Karolina Hansen, Michał Wypych.

**Investigation:** Justyna Olko, Michał Wypych, Olga Kuzawińska, Macéj Bańdur.

**Methodology:** Justyna Olko, Karolina Hansen, Michał Wypych.

**Project administration:** Justyna Olko.

**Supervision:** Justyna Olko.

**Validation:** Michał Wypych.

**Visualization:** Karolina Hansen, Michał Wypych, Olga Kuzawińska.

**Writing – original draft:** Justyna Olko, Karolina Hansen, Michał Wypych, Olga Kuzawińska.

**Writing – review & editing:** Justyna Olko, Karolina Hansen, Michał Wypych, Macéj Bańdur.

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
