## [Decision Letter · Decision Letter 0]

2 Jun 2020

PONE-D-20-04867

From Discouragement to Empowerment. Insights from an ethnolinguistic vitality survey among the Kashubs in Poland

PLOS ONE

Dear Dr. Olko,

Thank you for submitting your manuscript to PLOS ONE. After careful consideration, we feel that it has merit but does not fully meet PLOS ONE’s publication criteria as it currently stands. Therefore, we invite you to submit a revised version of the manuscript that addresses the points raised during the review process.

We look forward to receiving your revised manuscript.

Kind regards,

Anastassia Zabrodskaja, Ph.D.

Academic Editor

PLOS ONE

Journal Requirements:

2. Please ensure that you refer to Figure 7 in your text as, if accepted, production will need this reference to link the reader to the figure.

3. Please upload a copy of Figure 8, to which you refer in your text on page 23. If the figure is no longer to be included as part of the submission please remove all reference to it within the text.

4. Please include a caption for figure 8.

Additional Editor Comments (if provided):

Reviewers' comments:

Reviewer's Responses to Questions

**Comments to the Author**

1. Is the manuscript technically sound, and do the data support the conclusions?

Reviewer #1: Yes

Reviewer #2: Yes

Reviewer #3: Yes

2. Has the statistical analysis been performed appropriately and rigorously? 

Reviewer #1: Yes

Reviewer #2: Yes

Reviewer #3: Yes

3. Have the authors made all data underlying the findings in their manuscript fully available?

Reviewer #1: Yes

Reviewer #2: Yes

Reviewer #3: Yes

4. Is the manuscript presented in an intelligible fashion and written in standard English?

Reviewer #1: Yes

Reviewer #2: Yes

Reviewer #3: Yes

5. Review Comments to the Author

Reviewer #1: I recommend the article for publication. The paper is clear and well-written, with well-chosen examples, argues for a good point.

I greatly enjoyed reading this paper, which is both intellectually stimulating and very carefully written. A minor suggestion for improvement could be: to possible strengthening of the argument in Discussion and conclusions, you might want to provide more links with similar studies conducted in other settings.

Reviewer #2: I think the article could be published as it stands. There are two minor points the authors might wish to consider to help the reader navigate through the article and also make the article slightly more nuanced.

(1) The introduction to the article could signpost more what is to come in the rest of the article, through a short framework/outline. At the moment, for example, the linguistic community is very briefly referred to right at the beginning and the reader - particularly if not familiar with the particularities of the Kashubian language community - is given only a few relevant details, but nothing substantial. If would be helpful if the authors could indicate that a more substantive description is to follow below, as this is not obvious from the outset. Ditto for the other points mentioned in the introduction. This framework need only take up a few sentences at the end of the introduction.

(2) A minor point but it would be worthwhile the authors considering adjusting the description of speakers of Kashubian, which at the moment is: "While there may indeed be as many as 100,000 active speakers nowadays, there are approximately several tens of thousands of people who know Kashubian but rarely speak it". There will be a whole range of people who engage with Kashubian on a variety of levels, not just 'active speaker - non-active speaker', such as described in "Speakers and communities”, by Colette Grinevald & Michel Bert, pp. 45-65 in The Cambridge Handbook of Endangered Languages (2011). A reference to a more developed speaker framework would add more substance to this section of the article.

Reviewer #3: The article is clear and cohesive in its exploration of ELV within the Kashubian language group in Poland. The data analysis conducted provides relevant and unique insight into the current situation and vitality of the language. However the article would however benefit from a more socio-political framing. The authors do not discuss the current situation in Poland with regard to language rights or why Kashubian is now perceived to have a high rate of vitality. If further context is provided at the beginning of the article, the significance of the results will increase.

There were also some minor spelling and grammar issues (eg. in figure 1 on y axis), as well as the referencing format being confusing. Why are footnotes used when they are not at the bottom of the page? This makes it difficult to ensure the article is supported by relevant and up to date theoretical frameworks and methodologies.

With some minor review I would recommend this paper for publishing as it contributes to current discussion of ethnoloiinguistic vitality and prominence of minority languages, however again, more social background would add strength to the study.

6. PLOS authors have the option to publish the peer review history of their article (what does this mean?). If published, this will include your full peer review and any attached files.

Reviewer #1: No

Reviewer #2: No

Reviewer #3: No

---

## [Author Response · Author response to Decision Letter 0]

21 Jul 2020

Firstly, we would like to thank the reviewers for all their comments and recommendations. We found all of them to be very helpful for improving our paper. We have implemented all of the recommendations and, additionally, made some further improvements that build on these recommendations. 

In full accordance with the recommendations of Reviewer #2 and Reviewer #3, we have broadened and depended the introduction to the Kashubian community in the initial section of the paper, focusing on its socio-political situation and status. We have also added more socio-political framing regarding the current situation in Poland in terms of the language rights of national and ethnic minorities, including the state’s attitude toward them and the implications of their current legal status. Contextualizing the findings in this way facilitates a deeper understanding of the results discussed later in the paper and their significance. We have also included more references to the most recent relevant publications, including a book published in 2020 that was unavailable when we prepared the first draft of the paper. At the end of the “Introduction” we have also added a short description of the general structure of the paper, as a sort of “roadmap” for readers.

In the section “Context of the study” we have added to the historical overview of the Kashubs, with a special focus on post-1945 policies toward the group, which are important for understanding their present situation. We have also provided a more concise description of the group’s situation and vitality after 1989, as well as the changes that were set in motion after the recognition of Kashubian as a regional language in 2005. This includes an up to date overview of current spaces of language use and development. Moreover, these amendments better contextualize and explain the 2011 census data as it relates to the Kashubs and the Kashubian language. In full accordance with the recommendation of Reviewer #2, we have also provided a description of different types of Kashubian speakers and their language proficiencies, based both on earlier research (with additional references included) and our own knowledge of the topic. The latter embraces the emic knowledge and experience of one of the co-authors, who is a Kashubian scholar, teacher, activist and publicist. We decided to complement this description on kinds of speakers and spaces of language use by deepening the presentation and discussion of the related quantitative data obtained in our survey and described in the section “Results”. This improvement offers sociolinguistic insight that is required for better understanding the profile of our survey participants and for interpreting the survey results. Thus, we have described with more precision and in more detail participants’ self-reported oral and written skills in Kashubian and Polish, along with their domains of language use, adding an additional Table 1 that illustrates our data.

Finally, as suggested by Reviewer #1, we have strengthened our arguments in the section “Discussion and conclusions” and we have provided more references to other similar or relevant studies. Thus, we have included more insights from additional studies on the Rejection-Identification Model, which is particularly relevant for our case study and our discussion; we have also added a substantial section on the concept of empowerment and previous approaches to the theme of empowerment of ethnic minorities and, particularly, of speakers of minority/indigenous/endangered languages. In accordance with this discussion and our theoretical framework, we have also decided to change empowerment to self-empowerment in the title and relevant parts of the paper, as it describes more adequately the hypothesis and the interpretation of findings developed in this study, thus strengthening our arguments. Finally, we have consolidated our arguments with regard to the profile of our survey participants who represent an active and engaged sector of the Kashubian community’. 

We have also corrected spelling and grammar issues in the figures accompanying the paper and we provide them in a much better quality in the form of TIFF files. We also changed the type of charts presented in Figures 2, 3 and 4 to stacked bar charts, which makes them easier to read. Regarding other formatting details, including references, we follow the guidelines of the journal. Finally, we have implemented other minor editing improvements throughout the paper. 

Summing up, we have fully followed the recommendations of the reviewers, with with which we fully agree. In addition to implementing their direct suggestions, we have also introduced some additional improvements to other parts of the paper that relate to these recommendations and contribute to the overall coherence of the paper, the strength of our arguments and the interpretation of the results. 

With kind regards,

Justyna Olko

---

## [Editor Report · Decision Letter 1]

27 Jul 2020

From Discouragement to Self-Empowerment. Insights from an ethnolinguistic vitality survey among the Kashubs in Poland

PONE-D-20-04867R1

Dear Dr. Olko,

We’re pleased to inform you that your manuscript has been judged scientifically suitable for publication and will be formally accepted for publication once it meets all outstanding technical requirements.

Kind regards,

Anastassia Zabrodskaja, Ph.D.

Guest Editor

PLOS ONE

---

## [Editor Report · Acceptance letter]

5 Aug 2020

PONE-D-20-04867R1 

From Discouragement to Self-Empowerment. Insights from an ethnolinguistic vitality survey among the Kashubs in Poland 

Dear Dr. Olko:

I'm pleased to inform you that your manuscript has been deemed suitable for publication in PLOS ONE. Congratulations! Your manuscript is now with our production department. 

Kind regards, 

on behalf of

Dr. Anastassia Zabrodskaja 

Guest Editor

PLOS ONE